# Navigating cross-reactivity and host species effects in a serological assay: A case study of the microscopic agglutination test for *Leptospira* serology

**Riley O. Mummah**[1☮]*, **Ana C. R. Gomez**[1☮], **Angela H. Guglielmino**[1],
**Benny Borremans**[1,2,3], **Renee L. Galloway**[4], **Katherine C. Prager**[1], **James O. Lloyd-Smith**[1]

**1** Department of Ecology and Evolutionary Biology, University of California, Los Angeles, California, United States of America, **2** Wildlife Health Ecology Research Organization, San Diego, California, United States of America, **3** Evolutionary Ecology Group, University of Antwerp, Antwerp, Belgium, **4** Bacterial Special Pathogens Branch, Centers for Disease Control and Prevention, Atlanta, Georgia, United States of America

☮ These authors contributed equally to this work.
* rom5173@ucla.edu

**Data Availability Statement:** All code and data required to reproduce the study can be found at: https://github.com/rileymummah/x-reactivity/.

## Abstract

### Background

Serology (the detection of antibodies formed by the host against an infecting pathogen) is frequently used to assess current infections and past exposure to specific pathogens. However, the presence of cross-reactivity among host antibodies in serological data makes it challenging to interpret the patterns and draw reliable conclusions about the infecting pathogen or strain.

### Methodology/Principal findings

In our study, we use microscopic agglutination test (MAT) serological data from three host species [California sea lion (*Zalophus californianus*), island fox (*Urocyon littoralis*), and island spotted skunk (*Spilogale gracilis*)] with confirmed infections to assess differences in cross-reactivity by host species and diagnostic laboratory. All host species are known to be infected with the same serovar of *Leptospira interrogans*. We find that absolute and relative antibody titer magnitudes vary systematically across host species and diagnostic laboratories. Despite being infected by the same *Leptospira* serovar, three host species exhibit different cross-reactivity profiles to a 5-serovar diagnostic panel. We also observe that the cross-reactive antibody titer against a non-infecting serovar can remain detectable after the antibody titer against the infecting serovar declines below detectable levels.

### Conclusions/Significance

Cross-reactivity in serological data makes interpretation difficult and can lead to common pitfalls. Our results show that the highest antibody titer is not a reliable indicator of infecting serovar and highlight an intriguing role of host species in shaping reactivity patterns. On the other side, seronegativity against a given serovar does not rule out that serovar as the

**Funding:** ROM, AHG, BB, KCP, JOL-S were supported by the Strategic Environmental Research and Development Program (SERDP, RC-2635) of the U.S. Department of Defense (https://serdp-estcp.mil/). KCP, BB, ACRG, JOL-S were supported by the U.S. National Science Foundation, Division of Environmental Biology (DEB-1557022; https://www.nsf.gov/), the USFWS American Rescue Plan Act (ARPA) Zoonotic Disease Initiative (F23AP00118-00; https://www.fws.gov/project/american-rescue-plan-act-zoonotic-disease-grant-program), the National Science Foundation, Division of Ocean Sciences (OCE-1335657; https://www.nsf.gov), and a Cooperative Ecosystem Studies Unit (CESU) Cooperative Agreement (#W9132T1920006; http://www.cesu.psu.edu). The funders had no role in study design, data collection and analysis, decision to publish, or preparation of the manuscript.

**Competing interests:** The authors have declared that no competing interests exist.

cause of infection. We show that titer magnitudes can be influenced by both host species and diagnostic laboratory, indicating that efforts to interpret absolute titers (e.g., as indicators of recent infection) must be calibrated to the system under study. Thus, we implore scientists and health officials using serological data for surveillance to interpret the data with caution.

## Author summary

Serology is frequently used for disease surveillance, especially in systems that are resource constrained or logistically challenging. Serological testing involves analyzing blood serum samples to detect antibodies with reactivity toward specific pathogens (or more generally, molecular antigens), with the goal of characterizing past exposure to those pathogens. However, these antibodies can be non-specific and may react against other related pathogens or strains–a phenomenon known as cross-reactivity. Interpretation of serological data exhibiting cross-reactivity is difficult and simplifying assumptions are often made (e.g., to interpret the strain that elicits the highest antibody titer level as the infecting pathogen strain). Our work shows that interpreting antibody data requires more nuance and more caution. Both absolute titers and relative reactivity against different strains can vary across host species and diagnostic laboratories, so it is essential to interpret these data in the appropriate context. These host species differences in antibody reactivity and cross-reactivity patterns make direct comparisons across species inadvisable.

## Introduction

Identification of current infections and past exposure to specific pathogens is fundamental to studying the epidemiology and ecology of infectious diseases. The correct identification of the infecting species or strain is the basis for understanding epidemiological linkages within and between host species. Serology, or the detection of serum antibodies formed by the host against an infecting pathogen, is used to detect individuals with current infections or prior exposure to a specific pathogen and is a widely used diagnostic for large-scale pathogen surveillance, particularly in wildlife systems.

Cross-reactivity among antibodies complicates serology-based surveillance of many pathogen groups including *Leptospira* spp., *Chlamydia* spp., *Shigella* spp., *Salmonella* spp., *Brucella* spp., rickettsiae, flaviviruses and hantaviruses [1–15]. There are at least three major challenges. First, in the absence of other evidence it is often assumed that the pathogen species or strain that elicits the highest antibody titer is the presumptive infecting agent. However, titer magnitudes can depend on many factors including host species, host immune history, laboratory reference strains, or time since infection, so cross-reactions can distort this picture. Second, absolute titers are used to estimate the recency of infection, but the quantitative titer dynamics (e.g. maximum titer values and the rates of titer decline) of cross-reacting antibodies may vary by pathogen strain or host species [15–17]. Thus, conclusions regarding the recency of infection for pathogens whose serological tests assess antibody titers against a panel of strains may differ depending on which antibody titer results are used, or on the relative strength of response across host species. Third, when rates of decline differ among strains, even the contrast between seronegative and seropositive results could be unreliable. Antibody titers against

the infecting strain could decline to undetectable levels while titers of cross-reacting antibodies against other strains may remain detectable.

Despite these challenges, serology has been the basis of classification schemes for some major pathogen groups, including pathogenic species of the genus *Leptospira*, which cause the globally important disease leptospirosis [18,19]. Historically, *Leptospira* has been classified into serovars based on serological reactivity, and further clustered into serogroups based on antigenic similarity among serovars [20]. In recent decades, genetic approaches have revealed that *Leptospira* serovar/serogroup classifications do not always align neatly with species delineations [20], and well-documented issues with cross-reactivity complicate the identification of serovars from serologic data alone [21]. Reliable identification to the serovar level can be achieved using pulsed-field gel electrophoresis (PFGE) or whole-genome sequencing if a culture isolate can be obtained and a reference strain library is available [22], but *Leptospira* is a slow-growing and fastidious organism, so culture is difficult. New genomic techniques have made it possible to acquire near-complete genome sequences without an isolate [23,24] or identify serovar with genetic determinants [25], but these are still cost-prohibitive and require specialized expertise, so serology remains widely used worldwide.

The microscopic agglutination test (MAT) is the serological diagnostic reference test for pathogenic species within the genus *Leptospira* [21]. The test consists of challenging serial dilutions of serum with live cultured bacteria and observing (with dark-field microscopy) the amount of agglutination that occurs due to serum antibodies binding to the antigen presented by the bacteria. For *Leptospira*, MAT is performed using a panel of cultured isolates representing different serovars; often 1–15 isolates are used for veterinary purposes, or more than 20 isolates for human medicine. Serovars are chosen for MAT panels based on what is known to circulate in the region or host species being tested. However, anti-*Leptospira* antibodies show a high degree of cross-reactivity in MAT results, whereby antibodies generated by infection with one serovar will react with antigens of multiple serovars, so it is common to obtain positive results against multiple serovars in the panel. Furthermore, paradoxical MAT reactions, in which the early response is directed most strongly to a non-infecting serovar, are common in humans and other host species and further complicate any effort to identify the infecting serovar from MAT results alone [15,17].

Unlike many commonly used serological tests, MAT does not require host-specific reagents, which facilitates direct comparison of results from multiple host species. This is potentially beneficial as many *Leptospira* serovars infect multiple mammal hosts, and there are often questions about whether infections in different host species arise from multi-host circulation of a shared serovar or from multiple co-circulating serovars. Correctly interpreting the differences and similarities in MAT results across different species is an important step in describing the ecology of *Leptospira* in a potential multi-host system. Yet the possibility that MAT reactivity or cross-reactivity differs across host species has not been investigated or characterized.

In practice, many epidemiological and ecological studies of leptospirosis rely only on serum MAT data due to its affordability, relative ease, and lack of reliance on obtaining isolates. MAT is recognized as unreliable for serotyping because of cross-reactivity among serovars, but as it is often the only evidence available, especially for wildlife systems, many authors use it as a basis to speculate on the infecting serovar in their systems (e.g., [26–30]). Similarly, MAT titers are often interpreted as markers of recent versus older exposure [16,31]. The *Leptospira* research community is promoting more cautious interpretation of MAT results, including that MAT panel data should be interpreted only as presumptive serogroup (not serovar) of the infecting agent, and that any quantitative comparisons should be based on at least a four-fold difference in titer [32]. Yet even these guidelines reflect a degree of pragmatism, so further evaluation of this widely-used tool is warranted.

In our study, we leverage a unique ecological system with one circulating serovar of *Leptospira interrogans* in three sympatric wildlife host species and test the reliability of MAT as a tool to infer epidemiological processes. We specifically investigate the interpretation of maximum titers as markers of infecting serovar and the interpretation of titers as markers of time since exposure. We also highlight the potential confounding of host species and laboratory effects. Our results suggest that all MAT results (i.e., both absolute and relative quantitative titers) should be interpreted with great caution and consideration of host species and other system-specific effects.

## Data & methods

### Ethics statement

All California sea lion samples were collected under authority of Marine Mammal Protection Act Permits No. 932-1905-00/MA-009526 and No. 932-1489-10 issued by the National Marine Fisheries Service (NMFS), NMFS Permit Numbers 17115–03, 16087–03, and 13430. The sample collection protocol was approved by the Institutional Animal Care and Use Committees (IACUC) of The Marine Mammal Center (Sausalito, CA; protocol # 2008–3) and the University of California Los Angeles (ARC # 2012-035-12. UCLA is accredited by the Association for Assessment and Accreditation of Laboratory Animal Care (AAALAC) International. The Marine Mammal Center and UCLA adhere to the national standards of the U.S. Public Health Service Policy on the Humane Care and Use of Laboratory Animals and the USDA Animal Welfare Act. Isoflurane gas was used to anesthetize all wild-caught, free-ranging sea lions for sampling. All island fox and skunk samples were collected by the National Park Service under USFWS permit TE-08267-2.

### Study animals and sample collection

Our dataset comprises samples from California sea lions (*Zalophus californianus*), island foxes (*Urocyon littoralis*), and island spotted skunks (*Spilogale gracilis*) with confirmed infections of *L. interrogans* serovar Pomona. Samples were collected from 107 sea lions that had stranded along the central California coast between 2004–2017 and were admitted to The Marine Mammal Center (TMMC; Sausalito, California) for rehabilitation. An additional thirty sea lion samples were collected from free-ranging wild sea lions from the central California coast and northern Oregon, between 2010 and 2012, as described in Prager et al., 2020 [33]. The majority of sea lions were diagnosed with acute leptospirosis (97/137) based on clinical signs, serum chemistry results, and necropsy data [34].

Samples from island foxes (n = 59) and island spotted skunks (n = 4) were collected between 2011 and 2016 during annual grid and target trapping conducted by the National Park Service (NPS) as part of a monitoring program on Santa Rosa Island, California. Santa Rosa Island has an area of approximately 214 km$^2$ and only three terrestrial mammal species (island foxes, island spotted skunks, and island deer mouse [*Peromyscus maniculatus*]), and has no known history of *Leptospira* circulation before our study. Fox and sea lion data include both sexes and all age classes. All four skunks were adult males.

### Sample analysis

All animals included in this study had real time polymerase chain reaction (rt-PCR) confirmed *Leptospira* DNA in urine or kidney tissue as described by Wu et al [35], and the infecting *Leptospira* serovar was confirmed as *L. interrogans* serovar Pomona using PFGE (Fig A in

S1 Text) as described previously by Galloway & Levett [22] on all cultured isolates ($N_{CSL}$ = 19, $N_{fox}$ = 11, $N_{skunk}$ = 2).

Serum samples were tested by microagglutination test (MAT) against a panel of five *Leptospira* serovars comprising *L. interrogans* serovars Pomona (serogroup Pomona), Autumnalis (serogroup Autumnalis), Djasiman (serogroup Djasiman), Bratislava (serogroup Australis), and Icterohaemorrhagiae (serogroup Icterohaemorrhagiae). Most of the samples included in this analysis were tested against more than five serovars (56 CSL samples and 7 fox samples were tested with a 20-serovar panel). We exclude tested serovars that yielded almost entirely negative or very low results for all host species, and serovars for which the overlap between tested samples was low among the host species. All titers used in the host species comparison were analyzed at the Centers for Disease Control and Prevention (CDC) in Atlanta, Georgia using MAT (as described in Prager et al [36]) and run to endpoint dilution. Titer results were log-transformed for ease of interpretation using the following formula: $\log_2(\text{titer}/100) + 1$, thus a titer of 1:100 = 1, 1:200 = 2, 1:400 = 3, etc. Titers reported as <1:100 are represented by 0.

In a separate analysis focusing on variability among laboratories, a subset of 46 fox sera were MAT analyzed at three reference laboratories using a 2-serovar panel (Pomona and Autumnalis). The laboratories are referred to as Laboratories A, B, and C. Antibody titers against serovar Pomona were evaluated to endpoint at all three laboratories. Serovar Autumnalis was not titrated to endpoint for all samples at all laboratories. At Laboratory A, 43 of 46 samples were titrated to endpoint and 3 of 46 were only tested at a dilution of 1:100 (all were positive). At Laboratory B, all 46 samples were titrated to endpoint. At Laboratory C, all 46 serum samples were titrated to a 1:6400 dilution ($\log_2$ titer = 7) but not beyond.

## Data selection and analysis

To analyze antibody cross-reactivity patterns within and between host species, we selected MAT results from animals for which there was at least one positive urine PCR or culture result, which confirms current *Leptospira* infection. We did separate analyses for animals with PCR- or culture-confirmed infection ($n_{CSL}$ = 137; $n_{fox}$ = 59; $n_{sku}$ = 4) and animals with confirmed infection and PFGE-confirmed serovar ($n_{CSL}$ = 19; $n_{fox}$ = 11; $n_{sku}$ = 2). Only two skunk samples were PFGE-positive, so we included samples from all four skunks in both analyses. We also performed an additional comparison of PCR- or culture-confirmed infections in skunks with all MAT-positive skunks to confirm that patterns were consistent. For individuals that had been sampled longitudinally, we selected the MAT result from the serum sample with a collection date closest to that of the positive urine sample. The majority of MAT results from foxes (55/59) and all from skunks (4/4) were from sera collected on the same day as the *Leptospira* PCR- or culture-positive urine. Sea lion serum samples used for MAT were collected within 5 days of the date that the PCR- or culture-positive urine or kidney sample was collected (range = 0–5 days, median = 0 days). To analyze relative titer magnitudes among host species, we standardized antibody titers by dividing a given antibody titer by the highest antibody titer detected against any serovar in the 5-serovar MAT panel for that host serum sample.

We tested differences in the titer profiles across species using a multivariate nonparametric analysis of similarities test (ANOSIM; [37]). We used non-parametric Kruskal-Wallis tests to evaluate differences in titer magnitude among groups (i.e., species or serovar; [38]). When appropriate, a Wilcoxon rank sum test with a Bonferroni correction was used to assess pairwise differences between groups.

We evaluated a subset of 46 fox serum samples at three certified testing laboratories as described above (see section on Sample Analysis) to compare MAT results across laboratories. Fox serum samples were chosen for this laboratory comparison based on MAT titer results

from Laboratory A. For each MAT antibody titer level ranging from 1:100–1:51200, three serum samples with that MAT antibody titer against serovar Pomona, as reported by Laboratory A, were selected where possible (Table A in S1 Text). In addition to these 30 samples, we included a further 10 samples that had no detectable antibodies against serovars Pomona and Autumnalis at Laboratory A, and six samples that had no detectable antibodies against serovar Pomona but were MAT positive against serovar Autumnalis at Laboratory A. We tested for differences among laboratories using a Kruskal-Wallis test and between laboratories using a pairwise Wilcoxon rank sum test.

## Results

All host species exhibited strong antibody cross-reactivity against the five *Leptospira* serovars included in the MAT panel (Figs 1 and B in S1 Text). There was strong statistical evidence that the titer profiles are different among host species (ANOSIM p-value = 0.001; all pairwise comparisons between host species were also significant (Table B in S1 Text). MAT titers varied significantly across serovars in all host species (Table C in S1 Text). Titers against serovars Icterohaemorrhagiae and Pomona were significantly different from the other serovars in sea lion samples, whereas all serovars except Pomona and Djasiman were significantly different from each other for Channel Island foxes (Table D in S1 Text). No pairwise comparisons were significant for island skunks.

The serovar against which the highest antibody titer was measured differed among the three host species (Figs 1 and B in S1 Text), despite that all were infected by *L. interrogans* serovar Pomona (Fig A in S1 Text). The highest antibody titers detected in the majority of California sea lion (89.8%) and spotted skunk (100%) samples were against serovar Pomona, but the highest antibody titer detected in Channel Island fox samples was most often against serovar Autumnalis (69.5%). We note that serovars Pomona and Autumnalis belong to different serogroups. In 14 out of 59 fox samples (23.7%), the titer against serovar Autumnalis was at least

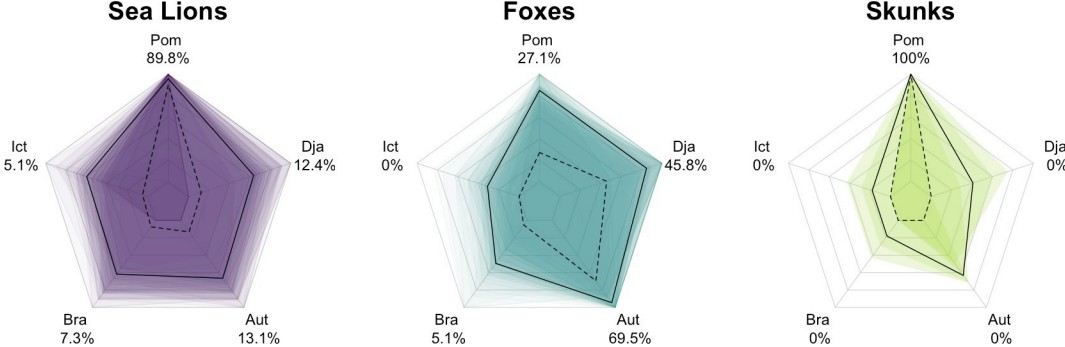

**Fig 1. Host-specific patterns of relative MAT antibody titers detected against five *Leptospira* serovars (Pomona, Djasiman, Autumnalis, Bratislava, and Icterohaemorrhagiae) when the infecting serovar is *L. interrogans* serovar Pomona.** Each plot shows the relative antibody titer levels (antibody titer against one serovar divided by the highest antibody titer detected against any serovar in the 5-serovar MAT panel run for that sample) for California sea lions (left; purple; n = 137), island foxes (middle; cyan; n = 59), and spotted skunks (right; green; n = 4). The shaded regions on each plot are a representative subsample of overlaid polygons for each species ($n_{CSL}$ = 59; $n_{fox}$ = 59; $n_{skunk}$ = 4), each linking the values for an individual sample. The continuous black line shows the relative antibody titer level for each sample (sample titer/maximum sample titer) averaged across all samples for each serovar for that species. The dashed black lines and the percentages associated with each serovar indicate the proportion of samples for which that serovar has the highest titer out of all serovars in that individual's panel, regardless of the actual titer. These numbers add up to more than 100% for sea lions and foxes, since multiple serovars can have the highest titer for any given sample (e.g., a particular individual could have highest titer of 1:6400 against both Pomona and Icterohaemorrhagiae).

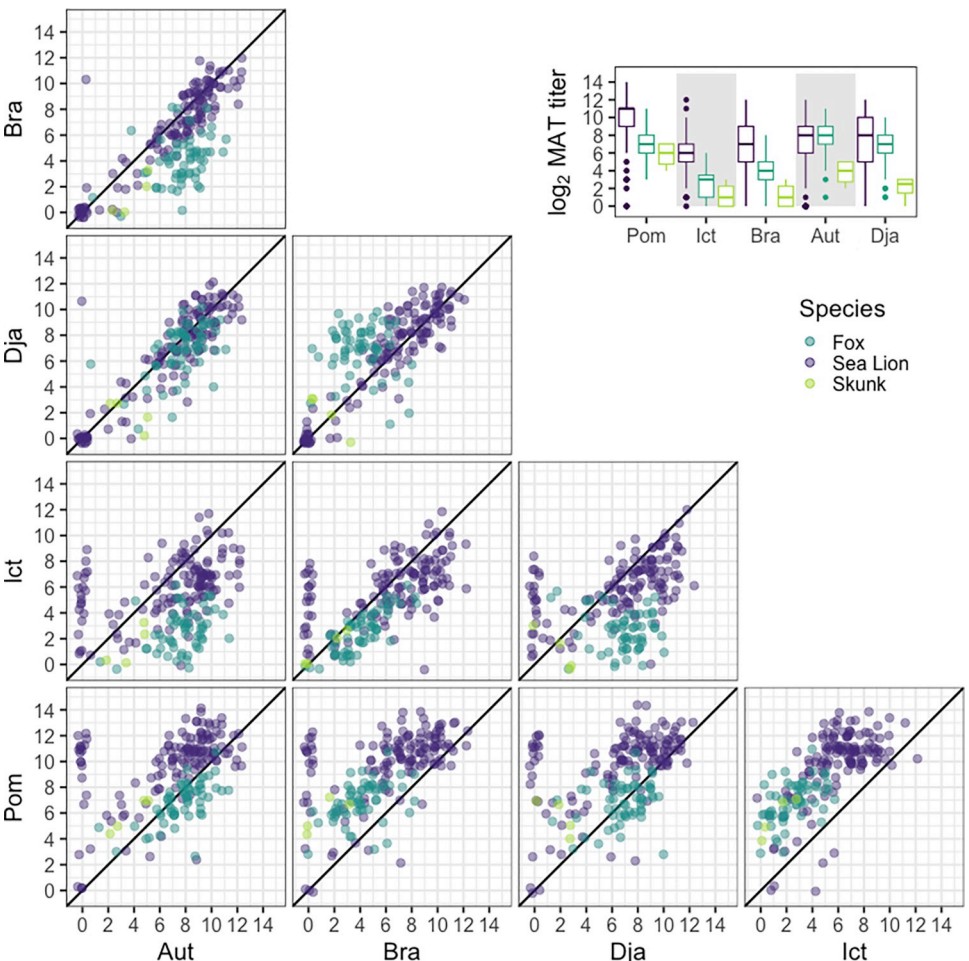

**Fig 2. Pairwise antibody titer levels against *Leptospira interrogans* serovars Pomona, Djasiman, Autumnalis, Bratislava, and Icterohaemorrhagiae in three host species.** Each plot shows the pairwise endpoint MAT titer levels (log₂ dilutions) for California sea lions (purple), island foxes (teal), and spotted skunks (green), all presumed to be infected with the same strain of serovar Pomona. The colors aggregate in a distinct pattern, showing that the serovar reactivity pattern is affected by the host species and that absolute titer magnitude differs among species. The black diagonal line corresponds to perfect equivalence between different serovars. Jitter has been added to the points to aid visualization. Inset: differences in MAT titer magnitude against each serovar among host species.

four-fold higher than that against serovar Pomona, i.e. beyond the conservative threshold for establishing a difference between two titer values.

We detected a clear difference in the absolute magnitude of anti-*Leptospira* antibody titers across the three host species (Figs 2 and C and Table E in S1 Text). Across four of the five serovars, sea lions exhibited consistently higher antibody titers relative to foxes and skunks (Fig 2). The exceptions were serovars Autumnalis and Djasiman, against which similar antibody titer magnitudes were detected in sea lions and foxes (Table F). Meanwhile, antibody titers detected in skunks were consistently lower than those from the other host species, though skunks and foxes shared similar antibody titer magnitudes for serovars Pomona and Icterohaemorrhagiae. Patterns were consistent between the PCR- and culture-confirmed dataset and the PFGE-confirmed dataset for all species (Figs A and C, and Tables G-J in S1 Text). Due to the small number of PCR-confirmed samples available for skunks, we further compared PCR- and culture-

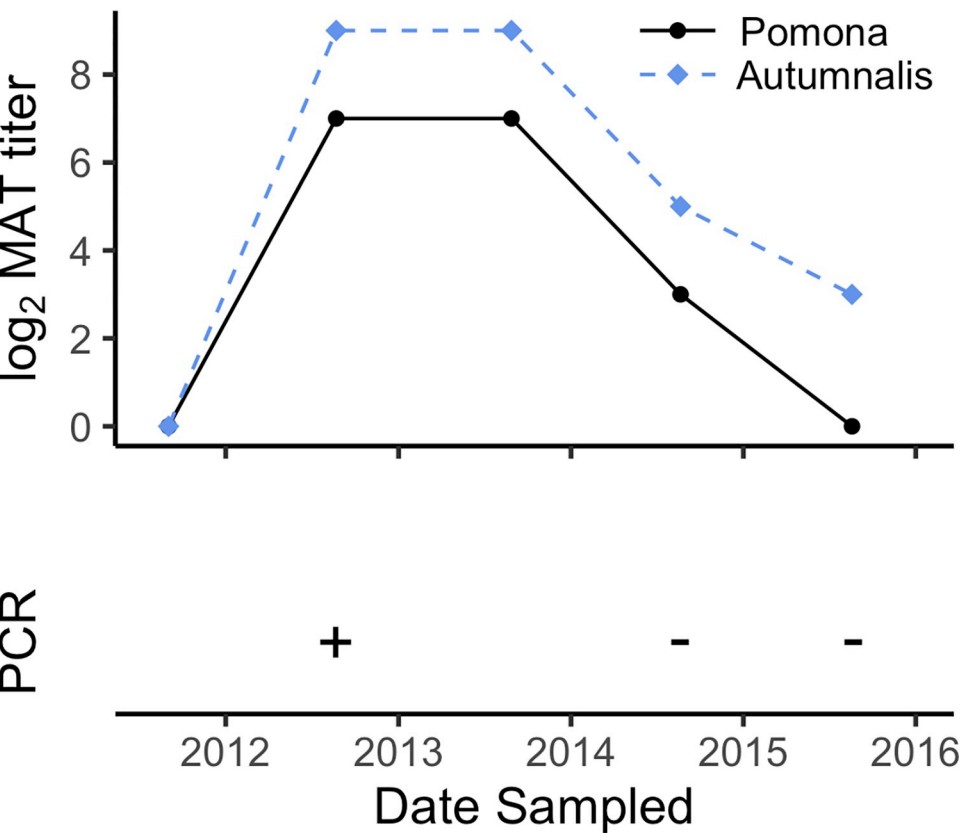

**Fig 3. Selected example of longitudinal antibody titer dynamics in a Channel Island fox infected by *L. interrogans* serovar Pomona.** The top panel shows antibody titers against *L. interrogans* serovars Pomona (black solid line) and Autumnalis (blue dashed line) from longitudinally collected serum samples from one fox. The bottom panel indicates the PCR test result from urine samples taken at the same time as serum collection.

confirmed skunks to all skunks that were MAT-positive against one of the five serovars on the panel and found similar results (Fig D in S1 Text; ANOSIM p-value = 0.356).

We examined titer dynamics and changes in the cross-reactivity profile through the course of infection and recovery using individual-level longitudinal data from 46 foxes sampled from 2009–2019. In particular, one fox illustrated a course of infection during which the titer against the non-infecting serovar (Autumnalis) was always higher than the titer of the infecting serovar (Pomona) and remained positive after the latter declined to undetectable levels (Fig 3). Although this was the clearest case study of this phenomenon in our dataset, other individuals had similar courses of infection where their highest titer was consistently against a non-infecting serovar (Fig E in S1 Text).

Analysis of 46 fox serum samples at three different diagnostic laboratories showed that both absolute and relative titers against serovars Pomona and Autumnalis varied systematically among laboratories (Fig 4A). The absolute antibody titer magnitude against serovar Pomona differed significantly amongst laboratories (p-value = 0.000348; Tables K and L in S1 Text). When comparing absolute antibody titer magnitude against serovar Pomona, the median titer was lowest from Laboratory A and highest from Laboratory C, with titers detected against serovar Pomona roughly one dilution greater at Laboratory B than Laboratory A, and more than three dilutions greater at Laboratory C than Laboratory A (Fig 4B). Endpoint titers against serovar Autumnalis were not run for all samples at all three laboratories, so comparisons were not possible at greater than

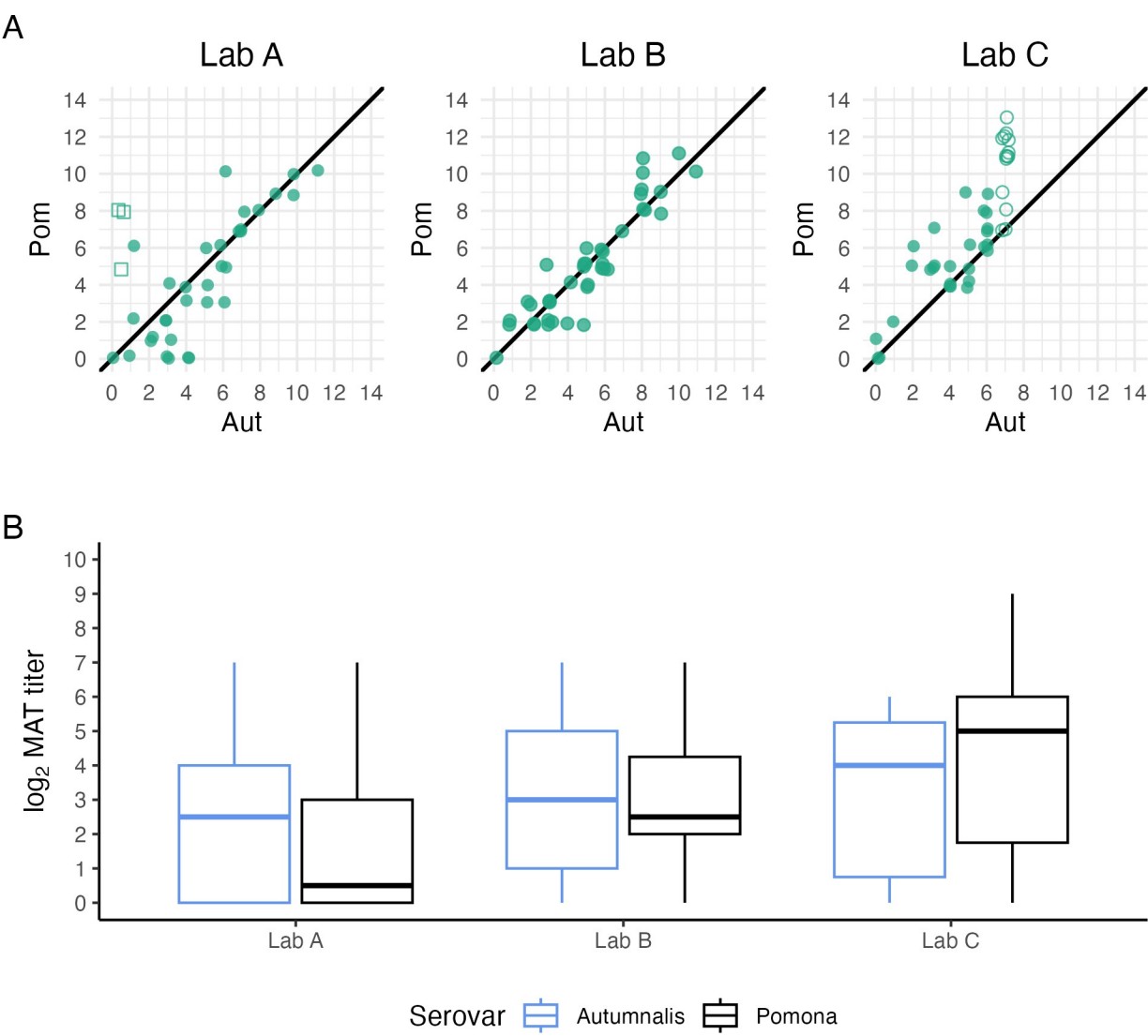

**Fig 4. Comparison of antibody titer results for fox serum samples evaluated at three testing laboratories.** Island fox serum samples (n = 46) were tested in three different certified testing laboratories. The MAT antibody titers ($\log_2$ dilutions) for serovars Pomona and Autumnalis are shown. All Pomona titers were run to endpoint dilution. In Panel A, open circles indicate non-endpoint Autumnalis titers at 1:6400 (log MAT titer 7) whereas open squares denote samples that were positive against serovar Autumnalis at 1:100, but no dilutions were performed. Jitter has been added to the points to aid visualization. Panel B represents the difference in antibody titer magnitude for a subset (n = 32) of samples that were run to endpoint for serovars Autumnalis and Pomona at all three laboratories.

1:6400 dilution ($\log_2$ titer = 7). Thirty-two of the samples tested at Laboratory C were positive at dilutions less than 1:6400 against serovar Autumnalis, but endpoint titers for the 14 samples that were still positive at the 1:6400 dilution are unknown. When assessing relative titer magnitude between laboratories, we found that at Laboratory A, antibody titers against serovar Autumnalis were generally higher than those against serovar Pomona (Fig 4), whereas at Labs B and C, antibody titers detected against serovar Autumnalis were generally equal to (Laboratory B) or less than (Laboratory C) those against serovar Pomona (Fig 4).

## Discussion

We tested sera from three host species at three different testing laboratories using the MAT assay and found that antibody cross-reactivity patterns can differ qualitatively and

quantitatively across host species, despite infection with the same causative agent (in our case study, the same species and serovar of *Leptospira*). We also showed that the highest detected antibody titer is not necessarily against the infecting serovar, and that both relative and absolute antibody titer magnitudes detected against different serovars can vary by diagnostic laboratory. MAT titers and cross-reactivity patterns are frequently used to characterize *Leptospira* epidemiology or ecology, with some studies proposing that the infecting serovar is that against which the highest MAT antibody titer is detected [26–30,39] or interpreting high MAT antibody titers against multiple serovars as evidence of multiple circulating strains [40]. Our results highlight that these interpretations are not robust [13,21] and the underappreciated role of host species in shaping MAT titers. This work raises clear caveats for the use and interpretation of MAT data, particularly in systems with multiple host species and can lead to inaccurate conclusions regarding the epidemiology of *Leptospira* transmission dynamics within and between host species. This work raises clear caveats for the use and interpretation of MAT data, as well as questions regarding the biological mechanisms by which host species can influence MAT results.

### Lesson 1: Highest titer does not always indicate infecting serovar or even serogroup

In our study, antibody titers detected in sea lions and skunks were generally highest against serovar Pomona, while foxes typically had the highest titer against serovar Autumnalis (Figs 1 and 2, and Figs A and B in S1 Text), despite our genetic evidence showing that the infections were caused by serovar Pomona. Notably, serovars Pomona and Autumnalis are in different serogroups [20]. These results highlight that the serovar against which the highest titer is detected should not be assumed to be the infecting serovar, and that even the more cautious approach of interpreting the highest titer as the presumptive serogroup of the infecting agent can lead to erroneous conclusions. Misidentification of the infecting serovar or serogroup could result in a misunderstanding of multi-species transmission patterns with implications for disease management and control.

### Lesson 2: Seronegativity must be interpreted with caution

Our longitudinal samples show that antibody titers against the infecting serovar can decay below the level of detection before those against non-infecting serovars do. Thus, a seronegative result against a given serovar does not necessarily mean it was not the infecting serovar, even when juxtaposed with positive titers against other serovars. This phenomenon could lead to misclassifying the infecting serovar if we rely on MAT for strain identification, or mistakenly ruling out the serovar that caused the infection, especially if exposure occurred in the relatively distant past.

### Lesson 3: Absolute and relative titer magnitudes depend on host species

We observed significant differences in both absolute and relative MAT titer magnitudes among the three host species tested. The same infecting serovar of *Leptospira* gave rise to different MAT cross-reactivity profiles in different host species (Fig 1). In general, we see that sea lions have higher median titers than foxes, which in turn have higher median titers than skunks across the five serovars (Fig 2). Autumnalis is a notable exception for which sea lions and foxes exhibit a similar median titer magnitude. The mechanisms underlying these differences are unknown.

## Lesson 4: Absolute and relative titer magnitudes can differ across laboratories

We observed systematic differences in absolute and relative titer magnitudes among three certified testing laboratories, including qualitative differences in which serovars elicited the highest titers from the same samples (Fig 4). Despite adherence to excellent laboratory standards and protocols, the nature of the MAT testing process means that some variation among laboratories is bound to exist. MAT is not standardized among laboratories, and variation both within and between laboratories is expected [41]. Many factors that are difficult to control can contribute to the variation of MAT results, so caution is needed when comparing MAT titers across laboratories. If such a comparison is necessary, ideally a set of samples should be run at both laboratories to calibrate the results.

## Implications for interpretation of MAT results

Of the more than 300 pathogenic *Leptospira* serovars currently described, most diagnostic MAT panels select a maximum of 20 serovars. In fact, cost and time restrictions typically limit panels to 4–6 serovars or fewer, particularly under conditions with fewer resources and lower testing capacity. This leaves the distinct possibility that a circulating serovar (and possibly the infecting serovar) could be omitted from the MAT panel, leading to potential for sub-optimal diagnostics and misunderstanding of circulating strains and transmission linkages. In other words, the serovar (or even serogroup) associated with the maximum titer in a given panel is not necessarily the infecting agent.

Titer magnitudes are often used to assess active infections. However, given our finding of differences in absolute titers across host species, relying on titer thresholds inferred from data in one species to identify recent or active infections in another can lead to inaccurate diagnosis and poor incidence estimates. For example, longitudinally sampled sea lions acutely infected with *L. interrogans* serovar Pomona had initial $\log_2$ titers against serovar Pomona ranging from 10 to 12 (i.e. 1:51200 to 1:102400) and these titers declined with a half-life of around 17 days [33]. Therefore if $\log_2$ titer thresholds used to define active infection in humans– 3 (or 1:400)–or dogs– 4 (or 1:800)–were applied to sea lions, many would be miscategorized as current infections [15,19]; this could occur even if the infecting serovar was not included in the MAT panel and the sea lion titers arose from antibody cross-reactivity. Our longitudinal fox data show that foxes could be similarly miscategorized if the human or dog thresholds are applied to them, as some foxes infected with *L. interrogans* serovar Pomona persist above the $\log_2$ titer = 4 threshold for years (Fig E in S1 Text). It is essential that any efforts to interpret absolute titers are calibrated to the system under study. When this is done, titer magnitudes (and their decay) can be used to estimate the recency of infection [16,42–44]. Modern titer kinetics approaches have the potential to include additional host-specific information about the relationship among serovars (i.e., the cross-reactivity profile of MAT titers) to estimate time since infection and improve our understanding of when outbreaks may have occurred.

We know of no prior work showing host species differences in MAT profiles. These patterns may be driven by different major histocompatibility complex (MHC) types and diversity [45,46], but more work is needed to understand how immunogenetic differences among wildlife may impact serology. It is noteworthy that the island fox population recently underwent a severe population bottleneck and exhibits very low genetic diversity (and therefore MHC diversity; [47]). Yet recent work in coyotes in southern California revealed a similar pattern– with MAT titers against serovar Autumnalis frequently exceeding those against serovar Pomona, despite known circulation of serovar Pomona in coyotes–suggesting that this effect may occur more broadly among some canids [48]. Systematically expanding surveillance

across canid species and beyond could provide insights on the possible existence of a host phylogenetic effect on MAT reactivity.

It is possible that some interspecies variation in titer magnitude was due to sampling bias. Over two-thirds of sea lion samples were from animals experiencing acute leptospirosis–the disease caused by *Leptospira* infection. By contrast, foxes and skunks were sampled during a routine trapping program aimed at monitoring these sensitive populations, so sample collection was not biased by disease severity. This could skew our observed antibody titers higher in the sea lions as their severe clinical disease suggests a recent infection [49], but a modeling analysis of island fox titers estimated peak titers against serovar Pomona of 6 to 9 on our $\log_2$ scale [16], consistent with values reported here for foxes, and lower than values reported for sea lions. It is clear that there is a large degree of immunological variability within and between species.

Variability in titer magnitudes has been documented across reference laboratories [50]. The International Leptospirosis Society sponsors the annual International Proficiency Testing Scheme for the Leptospirosis MAT, intended to provide information on the quality of MAT testing and improve MAT testing performance worldwide [51]. Early rounds of this program reported a wide variety of titers for the same sample and serovar [51]. Although multifactorial, variation is probably driven chiefly by two main factors. First, MAT relies on live bacterial cultures, and there may be slight strain variations among laboratories and among different batches grown within a laboratory. For trustworthy MAT results, within-culture serovar identity must be verified regularly [33,51]. Secondly, determining antibody titers by assessing agglutination under dark-field microscopy is subjective and requires significant expertise; even with best practices, some observer effect is inevitable. Altogether, many factors that are difficult to control can contribute to the variation of MAT results, so caution is needed when comparing MAT titers across laboratories. Ultimately, a difference of one serial dilution may not be a true difference at all given that convalescent testing requires a 4-fold rise to be deemed significant [32].

Overinterpretation of individual titer values can lead to misrepresentation of host relationships and circulating strains. This begs the question: given that genetic methods can provide clearer classification of pathogen strains and are becoming more accessible, why use MAT at all? Serology has many benefits that are distinct from culture-based methods with respect to duration of positivity and the potential to learn from antibody titer kinetics (with appropriately cautious interpretation). Despite issues with serological cross-reactivity and requirement of specialized laboratory and trained personnel, MAT is generally more affordable and sensitive than PCR or culture-based methods, is often an easier sample to collect, and captures information on past infections. Acute and convalescent MAT is also considerably more sensitive than organism-detection tests. Given the broad accessibility and continuing worldwide use of this diagnostic, we need to interpret its results with appropriate caution, while capitalizing on all available information. There may be an opportunity to improve assessment of the infecting serovar by exploiting consistent patterns in cross-reactivity against serovars within a host species, but more research is needed to describe these patterns within and across host species. The rising availability and falling cost of genetic methods, coupled with exciting new developments in obtaining whole genome sequences of *Leptospira* without culture isolates, point to a future where genetic typing adds clarity and certainty to *Leptospira* epidemiology and ecology.

## Conclusions

Serology plays an irreplaceable role in infectious disease ecology and epidemiology, but cross-reactivity can lead to pitfalls in interpreting serological data to assess current and past exposure

to specific pathogens. For our case study, we have shown that there can be substantial and consistent effects of host species that influence cross-reactivity profiles and quantitative titers, which could lead to erroneous conclusions about infecting serovars or recency of infection if appropriate caveats are not observed. This in turn can yield misleading interpretations about patterns of *Leptospira* circulation across host communities, or sources of zoonotic cases. This is especially true when relying on titer magnitude to determine infecting strain, or when samples have been analyzed at multiple laboratories. These findings have implications for all pathogens for which antibodies can cross-react with other species or strains, and we advise scientists and health officials using serological data for surveillance to interpret the data with suitable caution.

## Supporting information

**S1 Text.** Fig A. **Representative pulsed field gel electrophoresis (PFGE) results for 24 individual hosts (13 California sea lions, 9 island foxes, and 2 island skunks).** Labels beginning with CSL correspond to stranded sea lions, and WCSL corresponds to wild-captured sea lions. Three reference patterns for *L. interrogans* serovars Autumnalis, Icterohaemorrhagiae, and Pomona variant Kennewicki are also included. By convention, PFGE patterns that differ by three or fewer bands are considered the same serovar (Tenover et al 1995); by this criterion, all samples tested in our study were classified as *L. interrogans* serovar Pomona variant Kennewicki. Fig B. **Host-specific patterns of relative MAT antibody titers detected against five *Leptospira* serovars (Pomona, Djasiman, Autumnalis, Bratislava, and Icterohaemorrhagiae) when the infecting serovar is *L. interrogans* serovar Pomona, for individuals with positive PFGE only.** This is a replica of Fig 1 in the main text, except here only individuals with PFGE results are included. Each plot shows the relative antibody titers (antibody titer against one serovar divided by the highest antibody titer detected against any serovar in the 5-serovar MAT panel run for that sample) for California sea lions (left; purple; n = 19), Channel Island foxes (middle; cyan; n = 11), and spotted skunks (right; green; n = 4). The shaded regions on each plot are a representative subsample of overlaid polygons, each linking the values for an individual sample. The continuous black line shows the relative antibody titer level for each sample (sample titer/maximum sample titer) averaged across all samples for each serovar for that species. The dashed black lines and the percentages associated with each serovar indicate the proportion of samples for which that serovar has the highest titer out of all serovars in that individual's panel, regardless of the actual titer. These numbers add up to more than 100% for sea lions and foxes, since multiple serovars can have the highest titer for any given sample (e.g., a particular individual could have highest titer of 1:6400 against both Pomona and Icterohaemorrhagiae). Fig C. **Pairwise antibody titers against *Leptospira interrogans* serovars Pomona, Djasiman, Autumnalis, Bratislava, and Icterohaemorrhagiae in three host species, for individuals which are PFGE positive only.** MAT titers are shown as $\log_2$ dilutions. Statistical differences are shown in Tables G-J in S1 Text. Fig D. **Patterns of relative MAT antibody titers detected against five *Leptospira* serovars when the infecting serovar is *L. interrogans* serovar Pomona, for MAT-positive skunks (n = 55) and PCR-positive skunks (n = 4).** Fig E. **Longitudinal antibody titer dynamics in Channel Island foxes.** Each facet illustrates the antibody dynamics of a single fox. The top panel of each facet shows antibody titers against *L. interrogans* serovars Pomona (black solid line) and Autumnalis (blue dashed line) from longitudinally collected serum samples. The bottom panel in each facet indicates the PCR test result from urine samples taken at the same time as serum collection. Table A. The number of samples chosen per serovar Pomona titer level at Laboratory A for inter-laboratory titer comparison. Table B. Pairwise ANOSIM statistics across host species. Bolded

values represent statistical significance at $\alpha$ = 0.05. Table C. Kruskal-Wallis test statistics for differences among serovar within host species. Bolded values represent statistical significance at $\alpha$ = 0.05. Table D. Pairwise differences between serovar within a host species. P-values are shown for each pairwise comparison. Bolded values represent statistical significance at $\alpha$ = 0.05. Table E. Kruskal-Wallis test statistics for differences among host species for a given serovar. Bolded values represent statistical significance at $\alpha$ = 0.05. Table F. Pairwise differences between host species for a given serovar. P-values are shown for each pairwise comparison. Bolded values represent statistical significance at $\alpha$ = 0.05. Table G. Kruskal-Wallis test statistics for differences among serovar within host species for PFGE positive individuals only. Bolded values represent statistical significance at $\alpha$ = 0.05. Table H. Pairwise differences between serovar within a host species for PFGE positive individuals only. P-values are shown for each pairwise comparison. Bolded values represent statistical significance at $\alpha$ = 0.05. Skunks are not shown in this table because they only have one PFGE-confirmed individual. Table I. Kruskal-Wallis test statistics for differences among host species for a given serovar. Bolded values represent statistical significance at $\alpha$ = 0.05. Table J. Pairwise differences between host species for a given serovar. P-values are shown for each pairwise comparison. Bolded values represent statistical significance at $\alpha$ = 0.05. Table K. Kruskal-Wallis test statistics across laboratories for serovars Pomona and Autumnalis. Bolded values represent statistical significance at $\alpha$ = 0.05. Table L. Pairwise differences between laboratories for serovar Pomona. P-values are shown for each pairwise comparison. Bolded values represent statistical significance at $\alpha$ = 0.05.
(DOCX)

## Acknowledgments

The content of the information does not necessarily reflect the position or the policy of the U. S. government, and no official endorsement should be inferred. We would like to thank the volunteers, veterinarians, biologists and staff from The Marine Mammal Center (Sausalito, CA), The Marine Mammal Care Center Los Angeles, the Alaska Fisheries Science Center's Marine Mammal Laboratory, Oregon and Washington Departments of Fish and Game, at the U.S. Navy Marine Mammal Program, the Año Nuevo State Park, the University of California Santa Cruz's Año Nuevo Reserve, and the National Park Service for their logistical support and assistance with sample and data collection for this study.

**CDC DISCLAIMER:** The findings and conclusions in this report are those of the author(s) and do not necessarily represent the official position of the Centers for Disease Control and Prevention

## Author Contributions

**Conceptualization:** Riley O. Mummah, Ana C. R. Gomez, Benny Borremans, Katherine C. Prager, James O. Lloyd-Smith.

**Data curation:** Riley O. Mummah, Ana C. R. Gomez, Angela H. Guglielmino, Renee L. Galloway.

**Formal analysis:** Riley O. Mummah, Ana C. R. Gomez, Angela H. Guglielmino, Renee L. Galloway.

**Visualization:** Riley O. Mummah, Ana C. R. Gomez, Angela H. Guglielmino.

**Writing – original draft:** Riley O. Mummah, Ana C. R. Gomez.

**Writing – review & editing:** Riley O. Mummah, Ana C. R. Gomez, Angela H. Guglielmino, Benny Borremans, Renee L. Galloway, Katherine C. Prager, James O. Lloyd-Smith.

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
