## [Decision Letter · Decision Letter 0]

31 Mar 2024

Dear Mummah ,

Thank you very much for submitting your manuscript "Navigating cross-reactivity and host species effects in a serological assay: A case study of the microscopic agglutination test for Leptospira serology" for consideration at PLOS Neglected Tropical Diseases. As with all papers reviewed by the journal, your manuscript was reviewed by members of the editorial board and by several independent reviewers. In light of the reviews (below this email), we would like to invite the resubmission of a significantly-revised version that takes into account the reviewers' comments. 

Your manuscript has been reviewed by two experts and a major revision is suggested. I have also read the manuscript and I agree with the revision suggestions. 

We cannot make any decision about publication until we have seen the revised manuscript and your response to the reviewers' comments. Your revised manuscript is also likely to be sent to reviewers for further evaluation.

Sincerely,

Yung-Fu Chang

Academic Editor

Stuart Blacksell

Section Editor

Reviewer's Responses to Questions

**Key Review Criteria Required for Acceptance?**

**Methods**

-Are the objectives of the study clearly articulated with a clear testable hypothesis stated?

-Is the study design appropriate to address the stated objectives?

-Is the population clearly described and appropriate for the hypothesis being tested?

-Is the sample size sufficient to ensure adequate power to address the hypothesis being tested?

-Were correct statistical analysis used to support conclusions?

-Are there concerns about ethical or regulatory requirements being met?

Reviewer #1: This study sought to compare patterns of MAT reactivity among three different wildlife host species infected with Leptospira interrogans serovar Pomona. MAT results were also compared across three different laboratories and longitudinally.

-Are the objectives of the study clearly articulated with a clear testable hypothesis stated? YES

-Is the study design appropriate to address the stated objectives? 

In general, the study design was appropriate although the study lacks any use of statistics to compare results. Therefore it is difficult to know the validity of the conclusions made regarding the existence of differences among groups.

-Is the population clearly described and appropriate for the hypothesis being tested? YES. 

-Is the sample size sufficient to ensure adequate power to address the hypothesis being tested? For sea lions and foxes, there were a large number of samples available. The small number of skunks (4) limits the power of the study. The lack of statistical analysis makes it difficult to make conclusions about this.

-Were correct statistical analysis used to support conclusions? No statistical analysis available

-Are there concerns about ethical or regulatory requirements being met? No

Reviewer #2: The purpose of this study is to highlight the limitations of using serology for diseases with high antigenic variability. The example of MAT and leptospirosis is a notable one because, historically, it has been used to attempt serotyping when no other methods are available. The data and case studies provided in this study contribute to the already known fact that MAT on serum samples should not be used to identify serovars. The major issue with the report is that authors elaborate extensively on strain typing which implies a purpose that MAT is not meant to have. Furthermore, the leptospirosis scientific community has been increasingly recognizing this and promoting a more cautions interpretation. For example, the MAT panel titers are interpreted at the serogroup level and not serovar. Even when reporting as the “serogroup with the highest titers”, this should be interpreted in the context of other titers (i.e., >= 4-fold higher), and reported as presumptive serogroup. Nevertheless, the data reported provide a unique opportunity to show patterns of MAT serogroup titers when there is a known infecting serovar.

**Results**

-Does the analysis presented match the analysis plan?

-Are the results clearly and completely presented?

-Are the figures (Tables, Images) of sufficient quality for clarity?

Reviewer #1: -Does the analysis presented match the analysis plan? The analysis does match the analysis plan. 

-Are the results clearly and completely presented? At times it is a little difficult to follow the writing and there is the opportunity to simplify the information provided in table format. 

-Are the figures (Tables, Images) of sufficient quality for clarity? The figures are clear, appropriate, and of sufficient quality. Statistical analysis is needed. Please also include figures that demonstrate the relatedness of the isolates by PFGE.

Reviewer #2: Line 107: “Positive antibody titers against different strains make assessment of the infecting serovar and identification of epidemiological linkages difficult“. This is misleading as we know it is not meant to be used for this serovar identification. It could be rephrased to say that commonly titers are interpreted to determine presumptive serogroup. 

Line 110: “complicate any effort to identify the infecting strain from MAT results alone”. Remove the language referring to “strain” since MAT interpretation is for serogroups. 

Line 122: “the strain can be typed reliably (or identified as a potential new strain) by PFGE or genome

sequence typing methods, which are much faster and cheaper alternatives to CAAT”. This is referring to strain typing which has a different dimension and purpose than serotyping. In general, the introduction needs reorganization to focus on the limitations of MAT for serogroup identification. It is not clear what the authors mean by “strain” in the context of MAT interpretation since they can only refer to serogroups. 

Line 128: explanation of serovars and serogroups should be earlier in the introduction.

**Conclusions**

-Are the conclusions supported by the data presented?

-Are the limitations of analysis clearly described?

-Do the authors discuss how these data can be helpful to advance our understanding of the topic under study?

-Is public health relevance addressed?

Reviewer #1: -Are the conclusions supported by the data presented? -Are the limitations of analysis clearly described?

In general, the data support the conclusions. The conclusions would be greatly strengthened by statistical comparisons. There is no discussion of the reason for not performing a statistical analysis. The authors should also acknowledge the fact that a difference of one serum dilution may not actually be a difference at all (hence the requirement for a 4-fold rise to show a significant difference when doing convalescent testing). 

-Do the authors discuss how these data can be helpful to advance our understanding of the topic under study? YES

-Is public health relevance addressed? YES

Reviewer #2: Line 326: “it’s important to recognize that the serovar associated with the maximum titer in a given panel is not necessarily the infecting strain” . this needs to be interpreted in the context of serogroup. In the paper in general, using “strain” should be avoided. 

Line 380: MAT still requires specialized laboratory and trained personnel and it is difficult to access in many parts of the world.

**Editorial and Data Presentation Modifications?**

Reviewer #1: The manuscript is reasonably well written, although conciseness could be improved. Some minor errors:

Line 79: “flavivruses” missing an “i”, rickettsia should be rickettsiae

Line 119: with the right conditions, it can be days for culture rather than months

Line 217: consider rewording “PCR- or culture-confirmed skunks” (they were not confirmed as skunks using those methods)

Line 259: labs should be laboratories (also throughout the manuscrupt)

Line 381: acute and convalescent MAT is considerably more sensitive than organism-detection tests, this is really the main reason for doing it

Reviewer #2: Citations. Entire text needs to be checked. Different parts have different formats.

**Summary and General Comments**

Reviewer #1: This was a really interesting study and unique opportunity to examine the utility of MAT for identification of the infecting serovar, and highlight the reasons why MAT does not accurately predict the infecting serovar. The inability of MAT to predict the infecting serovar has long been known, but despite this, clinicians and researchers continue to try to use it for this purpose. The primary reason for major revision is the need for statistics to support the conclusion, more data to convince the reader of the relatedness of isolates (PFGE figures), and also to address the length of the manuscript. The introduction and discussion could be shortened substantially (e.g., half the length) as they reiterate what is already widely reported in the literature. I would also recommend including detail in the abstract regarding which wildlife species were included.

Reviewer #2: (No Response)

PLOS authors have the option to publish the peer review history of their article (what does this mean?). If published, this will include your full peer review and any attached files.

Reviewer #1: No

Reviewer #2: No
---

## [Decision Letter · Decision Letter 1]

8 Aug 2024

Dear Mummah,

Thank you very much for submitting your manuscript "Navigating cross-reactivity and host species effects in a serological assay: A case study of the microscopic agglutination test for Leptospira serology" for consideration at PLOS Neglected Tropical Diseases. As with all papers reviewed by the journal, your manuscript was reviewed by members of the editorial board and by several independent reviewers. The reviewers appreciated the attention to an important topic. Based on the reviews, we are likely to accept this manuscript for publication, providing that you modify the manuscript according to the review recommendations. 

Sincerely,

Yung-Fu Chang

Academic Editor

Stuart Blacksell

Section Editor

Reviewer's Responses to Questions

**Key Review Criteria Required for Acceptance?**

**Methods**

-Are the objectives of the study clearly articulated with a clear testable hypothesis stated?

-Is the study design appropriate to address the stated objectives?

-Is the population clearly described and appropriate for the hypothesis being tested?

-Is the sample size sufficient to ensure adequate power to address the hypothesis being tested?

-Were correct statistical analysis used to support conclusions?

-Are there concerns about ethical or regulatory requirements being met?

Reviewer #1: (No Response)

Reviewer #2: The revised version addressed my concerns.

**Results**

-Does the analysis presented match the analysis plan?

-Are the results clearly and completely presented?

-Are the figures (Tables, Images) of sufficient quality for clarity?

Reviewer #1: (No Response)

Reviewer #2: No concerns

**Conclusions**

-Are the conclusions supported by the data presented?

-Are the limitations of analysis clearly described?

-Do the authors discuss how these data can be helpful to advance our understanding of the topic under study?

-Is public health relevance addressed?

Reviewer #1: (No Response)

Reviewer #2: The introduction and discussions are much more focused and all the concerns about MAT use and interpretation have been incorporated.

**Editorial and Data Presentation Modifications?**

Reviewer #1: (No Response)

Reviewer #2: (No Response)

**Summary and General Comments**

Reviewer #1: The authors have improved this manuscript, although I think that there is still room to make it more concise. My line references are for the version of the manuscript with track changes visible (all markup).

Line 170 “reliable…can be achieved using PFGE or WGS” needs a reference. Serovar cannot be directly inferred from PFGE and WGS information, a reference strain library that correlates sequence type back to serovar is required (as in the new figure).

Line 181. For veterinary purposes, often 1 to 15 cultured isolates are used; in human medicine over 20 serovars may be included in MAT (the authors actually state this on line 439)

Line 200. “quantitative MAT titer levels” – could be just “MAT titers,” also elsewhere in document

Line 395-396. I think this sentence needs to be deleted as it is redundant.

Line 443. “At the bottom line…” This sentence is not concise and could even be deleted.

Line 543. “Genetic methods remain superior to serology for serotyping…” does not make sense. By definition, serotyping is a serologic technique. The benefit of MAT is that it is more sensitive for diagnosis that organism-detection tests such as PCR and culture; this needs to be stated.

The new supplemental figure improves the manuscript.

Reviewer #2: (No Response)

PLOS authors have the option to publish the peer review history of their article (what does this mean?). If published, this will include your full peer review and any attached files.

Reviewer #1: No

Reviewer #2: No

Figure Files:

Data Requirements:

Reproducibility:

References

---

## [Editor Report · Decision Letter 2]

4 Sep 2024

Dear Dr . Mummah,

We are pleased to inform you that your manuscript 'Navigating cross-reactivity and host species effects in a serological assay: A case study of the microscopic agglutination test for Leptospira serology' has been provisionally accepted for publication in PLOS Neglected Tropical Diseases.

Best regards,

Yung-Fu Chang

Academic Editor

Stuart Blacksell

Section Editor

---

## [Editor Report · Acceptance letter]

28 Sep 2024

Dear Mummah,

We are delighted to inform you that your manuscript, "Navigating cross-reactivity and host species effects in a serological assay: A case study of the microscopic agglutination test for Leptospira serology," has been formally accepted for publication in PLOS Neglected Tropical Diseases.

Best regards,

Shaden Kamhawi

co-Editor-in-Chief

Paul Brindley

co-Editor-in-Chief
